# Immigrant Older Adults’ Experiences of Aging in Place and Their Neighborhoods: A Qualitative Systematic Review

**DOI:** 10.3390/ijerph21070904

**Published:** 2024-07-10

**Authors:** Alesia Au, Sadaf Murad-Kassam, Vestine Mukanoheli, Sobia Idrees, Esra Ben Mabrouk, Khadija Abdi, Megan Kennedy, Kyle Whitfield, Jordana Salma

**Affiliations:** 1Faculty of Nursing, College of Health Sciences, University of Alberta, Edmonton, AB T6G 1C9, Canada; smurad@ualberta.ca (S.M.-K.); mukanohe@ualberta.ca (V.M.); sidrees@ualberta.ca (S.I.); ebenmabr@ualberta.ca (E.B.M.); kabdi@ualberta.ca (K.A.); sjordana@ualberta.ca (J.S.); 2Geoffrey & Robyn Sperber Health Sciences Library, University of Alberta, Edmonton, AB T6G 1C9, Canada; mrkenned@ualberta.ca; 3Faculty of Science, School of Urban and Regional Planning, University of Alberta, Edmonton, AB T6G 2R3, Canada; kw16@ualberta.ca

**Keywords:** immigrant, neighborhood, older adults, aging in place, social cohesion, ethnic enclave

## Abstract

Engaging in one’s neighborhood fosters independence, promotes social connectedness, improves quality of life, and increases life expectancy in older adults. There is a lack of evidence synthesis on immigrant older adults’ neighborhood perceptions and experiences, essential for addressing neighborhood-level influences on aging in place. This study systematically synthesizes qualitative evidence on immigrant older adults’ perceptions and experiences of their neighborhoods. A comprehensive search was conducted from inception to 5 April 2023, in multiple databases. This review considered studies including immigrant older adults aged ≥60 years, included studies from any country where the neighborhood was the focus, and only considered qualitative data while excluding review studies, theoretical publications, and protocols. Eligible studies were appraised using the JBI critical appraisal checklist for qualitative research. The Joanna Briggs Institute meta-aggregation approach was used to synthesize findings, and the ConQual approach established confidence in the synthesis. A total of 30 studies were included. Most studies were conducted in North America and explored phenomena such as aging in place, social capital, social cohesion, sense of community, and life satisfaction. Key contextual factors were walkable safe access to social spaces, accessible transportation to amenities, social cohesion with neighbors, and pre-migration neighborhood experiences. Immigrant older adults have varied experiences related to their sense of belonging and social cohesion. Factors such as racial discrimination, feeling unsafe, and social isolation contributed to negative perceptions. This review highlights the need for inclusive neighborhoods that align with the needs and values of immigrant older adults aging in place.

## 1. Introduction

The global focus on facilitating “aging in place” for the world’s growing aging population has increasingly permeated initiatives and responsibilities, trickling down from national policy agendas to local levels [1,2]. This is exemplified by initiatives such as Age Friendly Cities and Communities [3], in which local communities shoulder the responsibility to create infrastructure that allows older adults to live at home and avoid institutionalization. Aging in place (AIP) refers to a plethora of concepts and definitions characterized in relation to spatial conditions, care experiences, and/or exercise of individual choice or self-determination [4,5]. We define AIP as the place-related condition where individuals are located in neighborhoods and in areas most familiar and comfortable to them. This definition expands beyond the idea that individuals want to grow old in their own homes, towards remaining in the current community or neighborhood while living in their choice of residence [6].

The physical and social features of neighborhoods influence older adults’ quality of life [7]. The interaction between older adults and their social environment either encourages, hinders, or deters their movement within the neighborhood [8]. Older adults may experience decreased mobility due to frailty and other factors as they age [9]. Exploring within and beyond one’s neighborhood promotes independence and enhances social connections [9], thereby enhancing quality of life and extending life expectancy among older adults [10]. However, immigrant older adults (IOAs) are frequently marginalized in this discourse. An immigrant is operationally defined as an individual born in a country other than their current country of permanent residence and who has legal permanent status or citizenship.

Canada’s demographic landscape is experiencing substantial transformations, marked by an aging population and increased rates of migration, with immigrants accounting for over a quarter of the total population [11]. Immigrant older adults who are AIP often have strong transnational connections which includes social support networks that extend beyond their neighborhood and country of residence [12] and this might influence perceptions of local connections and supports. IOAs who are racially and linguistically different from mainstream local communities encounter exclusionary practices such as racism and discrimination, which restrict their access to neighborhood spaces that allow for social connectivity and belonging [13,14]. Social isolation and loneliness result in harmful mental and physical health consequences for older adults [15]. Furthermore, IOAs, especially newcomers, may experience limitations in walking in their neighborhoods due to language barriers, unfamiliarity with new environments, and discrimination [16]. Older immigrants often experience social exclusion and reduced participation in physical and social activities and face challenges such as language limitations and transportation barriers that limit their mobility and integration into their new place of residence [17,18].

Different cultural backgrounds and migration experiences can influence their perceptions of neighborhood environments, including safety, social cohesion, and available amenities [19]. Ethno-specific differences in the perceptions of barriers to walkability in the neighborhood and social connectedness is an important consideration for the development of appropriate and effective strategies for AIP [20]. The literature presents conflicting information from multiple studies, highlighting both the advantages and drawbacks of aging in ethnic enclaves or residing in multigenerational households [21,22,23,24,25]. Specifically, IOAs may encounter isolation if living away from ethnic enclaves and/or multigenerational households, yet their perceptions of quality of life vary.

In reality, the concept of a neighborhood encompasses a multifaceted social and spatial construct, presenting a challenge in establishing clear, significant, and universally accepted boundaries [26]. Research consistently shows that social determinants of health are strongly patterned by place [8]. Thus, to address the neighborhood or community contexts that affect older adults’ health across their life course, it is crucial to place at the center older adults’ own self-defined neighborhoods through activities, social ties, place attachments, and perceptions. Not only will this provide valuable insights for urban planners to understand IOAs’ motivations and how they navigate constraints related to leisure and social activities, but considering their preferences can enhance the likelihood of their participation and utilization of these planned spaces [27]. Expanding beyond objective indicators of neighborhoods such as ample greenspace, accessible amenities, and scores of safety, it is important to explore the perceptions of living in neighborhood settings as perceptions are correlated with subjective health outcomes [28,29].

A preliminary search of PROSPERO, MEDLINE, the Cochrane Database of Systematic Reviews, and JBI Evidence Synthesis was conducted and no current or in-progress systematic reviews on the topic were identified. To date, there is no comprehensive overview of the neighborhood experiences and perceptions of IOAs. AIP theory and practices heavily draw from what is known about aging in non-immigrant communities [30]. There is a notable research gap of the experiences and needs of IOAs in their neighborhood environments, so that researchers, policymakers, and community knowledge users can develop targeted strategies to promote integration, access to resources, and participation in physical and social activities. The aim of this review was to synthesize the experiences and perceptions of neighborhoods in IOAs.

### Review Question

How do IOAs experience the neighborhoods in which they live?

## 2. Methods

This systematic review was conducted in accordance with the JBI methodology for systematic reviews of qualitative evidence [31]. The Enhancing Transparency in Reporting the Synthesis of Qualitative Research (ENTREQ) guidelines were used to report the synthesis. The search results were tracked and reported using the Preferred Reporting Items for Systematic Reviews and Meta-Analysis (PRISMA). The protocol has been registered with PROSPERO before starting the review (Systematic review registration number: PROSPERO ID # CRD42023430767.).

### 2.1. Inclusion Criteria

We used a Population (Phenomena of) Interest, Context (PICo) framework to identify eligible studies [32].

This review considered studies that included IOAs aged ≥ 60 years and belonging to an immigrant group. If a study recruited participants from a non-immigrant group as a sub-sample, we included findings only from the immigrant sample if reported separately. We did not include studies only examining service or care providers’ experiences because our research question aimed to include experiences from the lens of older immigrants. We also did not include studies that reported participants in institutionalized care or studies considering institutional care facilities within neighborhoods (i.e., long-term care facilities and retirement care facilities). This review included studies from any country. Studies were only included if the neighborhood, defined loosely as the geographical space in proximity to where older adults live, was the focus of the study. No restrictions were placed on publication dates. All studies with a focus on qualitative data were considered, including research designs such as phenomenology, ethnography, grounded theory, action research, participatory research, and qualitative surveys. Mixed-method designs were considered with the condition that only qualitative data would be extracted. We excluded all types of review studies, discussion papers, theoretical publications, conference proceedings, protocols, and abstracts. We included grey literature to mitigate publication bias.

### 2.2. Search Strategy

The search strategy aimed to locate both published and unpublished studies. This review is reported following the PRISMA for Search (PRISMA-S) extension. A systematic literature search was conducted by a health sciences librarian who is experienced with synthesis review searching. Searches were performed in the following bibliographic databases from inception to 5 April 2023: Medline (1946—present), EMBASE (1974—present), PsycINFO (1806—present) via OVID, CINAHL (1936—present) and SocINDEX (1895—present) via EBSCOhost, Scopus (1976—present), Cochrane Library (1993—present) via Wiley, and Sociological Abstracts (1952—present) via ProQuest. Databases were searched using a combination of natural language keywords and controlled terms (subject headings), wherever they were available. Search concepts are as follows: (1) immigrants; (2) older adults; (3) neighbourhood or community perceptions/characteristics including AIP. No publication date or study type filters were applied to increase search sensitivity. The full search strategy appears in a Appendix A (Appendix A: Full Search Strategy). Results were exported from the databases in complete batches and imported into the synthesis review software, Covidence (Veritas Health Innovation, Melbourne, Australia, 2023), to manage the review results and facilitate screening and deduplication. Hand-searching the reference list of the included studies was completed to capture relevant studies that were not captured through searches. Using the same search concepts, grey literature was searched for through theses and dissertations via ProQuest, and policy briefs and reports via the first 200 results in Google.

### 2.3. Study Selection

A team-based approach (AA, KA, EBM, SI) was used for data screening with weekly team meetings. Covidence was used for title and abstract screening in stage one and for full text screening in stage two by two independent reviewers. A third reviewer helped reconcile discrepancies in team meetings. All the studies that met the inclusion criteria were moved to the data extraction stage.

### 2.4. Assessment of Methodological Quality

Continuing with a team-based approach (AA, SM, VM), eligible studies were critically appraised by two independent reviewers for methodological quality using the standard JBI critical appraisal checklist for qualitative research [33]. Each study was assessed using the JBI checklist, consisting of 10 questions, with one point awarded for each “Yes” response; studies receiving fewer than six “Yes” responses were excluded due to poor quality. Studies that had no report related to the question being appraised for were marked as “No”, while studies that had vague mention related to the question, or the information was missing were marked as “Unclear”. The results for each study were reported in a tabular format (Table 1). Upon completion of each study assessment by both reviewers, the results were compared between the two reviewers. In the case of non-consensus between the two reviewers, a third reviewer was used to make the final decision about the score of each study.

### 2.5. Data Extraction and Meta-Aggregation

A standardized data extraction format, derived from the modified JBI Qualitative Assessment Review Instrument (JBI QARI) data extraction tool [63], was used to extract the characteristics of each included study by two reviewers (KA, EBM), and reported in the tabular format. Information about study characteristics include the methodology used for data collection and analysis, phenomena of interest, context/setting of the study, number of participants, and type of population [minority group identified].

Three reviewers (AA, SM, VM) then worked to extract findings and their illustration, as reported in the included studies relevant to how a neighborhood was perceived or experienced by immigrant older adults (Appendix A: Extraction from Included Studies). The JBI ConQual process was used to evaluate the dependability and credibility of each synthesized finding [64]. Dependability was assessed using five items from the JBI critical appraisal checklist. These included congruence between the research methodology and the research question/objectives; the data collection methods; the representation and data analysis; statements locating the researcher culturally/theoretically; and addressing the influence of the researcher on the research and vice-versa. Dependability was rated high if 4–5 dependability questions were answered “Yes”, moderate if 2–3 dependability questions were answered “Yes”, and low if 0–1 dependability questions were answered “Yes”. The credibility was assessed in conjunction with the meta-aggregation process using a series of steps: (1) The participants’ quotations or author observations of context (illustrations) and the authors’ statements (findings) relevant to our aim were extracted by one reviewer (AA, SM, VM) and double-checked by another reviewer (AA, SM, VM). All extractions were selected from the results section only. (2) The two reviewers independently rated the credibility of each author statement to represent congruity between the findings and illustrations. There were three levels of credibility: unequivocal, credible, and unsupported (Table 2). Ongoing team meetings (AA, SM, VM, JS) were conducted to ensure a uniform approach was used across all included studies. (3) Findings rated as unequivocal or credible were grouped into categories. This involved the aggregation of findings to generate a set of statements that represented that aggregation, through pooling these findings on the basis of similarity in meaning. Any findings rated as unsupported were not included in further analysis. Each team member grouped findings from a sample of articles into categories independently. (4) The team worked together to compare categories and reach a consensus to create synthesized findings, which involved collating key categories and writing narrative descriptions of the interrelated categories.

### 2.6. Assessing Confidence in the Findings

The final synthesized findings were graded according to the ConQual approach for establishing confidence in the output of qualitative research synthesis and presented in a ConQual Summary of Findings [64]. An overall ConQual score, rated as high, moderate, low, or very low, was calculated for each synthesized finding. Using established guidelines, each finding started with a high rating and was downgraded a category for every downgrade in the dependability and credibility scores.

## 3. Results

### 3.1. Study Inclusion

The search strategy applied to digital databases searching yielded 4832 studies. After deduplication, two-stage screening and assessing eligibility, 16 studies were included in the review (Figure 1). The reasons for exclusion of studies are presented in the PRISMA. An additional 14 records were included from grey literature searching and hand-searching the reference lists of the 16 studies included from the database search. A total of 30 studies and records were included in the review.

### 3.2. Characteristics of Included Studies

Out of the 30 studies examined, 10 were conducted in America, 10 in Canada, 7 in Britain or Europe, and 3 in Australia or New Zealand. These studies varied in how they categorized their participants, focusing on the countries of origin, ethnicity, and/or native language. Among them, ten studies specifically involved older Chinese immigrants, while the remaining immigrant groups hailed from South Asian, Asian, Arab, Black Caribbean, Latino, Russian backgrounds, or were not specified. All studies utilized interviews as a data collection method, with six studies including group interviews. Additionally, other forms of methods included participant observations (*n* = 2), mapping interviews (*n* = 2), and photovoice (*n* = 2). The phenomena of interests across the included studies included AIP, social capital, social cohesion, social connection, sense of community, neighborhood friendliness, experiences of “place” or “home”, and life satisfaction. The full description of the study characteristics is summarized in Table 3.

### 3.3. Methodological Quality

The methodological quality of the 30 studies is summarized in Table 3. In total, 24 of the 30 studies received 8 “Yes” responses, with 9 studies that met the criteria 100% of the time. The remaining six studies received six or seven “Yes” responses. Overall, the methodological quality of the 30 eligible studies was considered moderate and no studies were excluded following critical appraisal.

The team aimed to be more inclusive by marking studies that did not explicitly state their philosophical perspective in the methodology as “Yes” for having a theoretical framework guiding their study. However, there still remained a relatively low congruity (*n* = 23) between philosophical perspectives and the research methodology. Nearly all studies found congruence between their research methodology and their research questions or objectives (*n* = 29), with their data collection methods (*n* = 28), with their representation, and analysis of the data (*n* = 30). The included studies had high congruity over the representation of participant voices (*n* = 29). Just over half of the included studies located the researcher culturally or theoretically (*n* = 19) and addressed the influence of the researcher on the research and vice-versa (*n* = 18). Finally, the conclusions of 26 of the included studies flowed from the analysis or interpretation of the data.

### 3.4. Review Findings

After data extraction and synthesis, four synthesized findings were identified (see Table 4 and Appendix A: Full Meta-Aggregation). A total of 19 categories and 243 findings were retrieved across the 30 studies. One of the categories (Lack of availability for senior housing in desired neighborhoods) was reported narratively as its meaning was not congruent with the other synthesized findings. Of the findings, 210 were rated as unequivocal and 33 were rated as credible. The confidence in the synthesis is presented in the ConQual Summary of Findings (Table 5).

### 3.5. Synthesized Finding 1

Walkability can be improved by enhancing safety features of neighborhoods to address physical disability limitations that are exacerbated by winter weather conditions. Preferences about living in areas with a warmer climate, clean spaces, and accessible greenspaces that allow for socialization were important factors for older immigrants to feel comfortable in their neighborhood. This synthesized finding was comprised of four categories.

#### 3.5.1. Category 1.1 Physical Accessibility and Walkability

IOAs described how the physical features of neighborhoods may pose challenges for their ability to safely move out of their home and within the neighborhood, particularly for those who experience physical disabilities. Comments on physical accessibility and ways to offset challenges posed by physical disabilities related to having accessible and maintained sidewalks, benches close by for resting, and attention to pedestrian safety.

*“The pavements and streets are very narrow and dangerous. People get hit by cars when they try to cross the road’ (71-year-old man)… There are no benches and there is hardly any green space around here’ (66-year-old woman).”* [38]

#### 3.5.2. Category 1.2 Winter Weather Challenges

Fear of falling was a common concern due to snow, ice, and having to wear bulkier clothing which led to avoiding outdoor activities in their neighborhoods in winter which in turn exacerbated social isolation. These concerns heighten their anxiety about burdening family members in the event of an accident. Navigating weather environments similar to their home countries was more feasible and this sometimes dictated choices around where they settled post-migration.

“*From our house to the bus stop, you have to walk over ice. [I] have to wear three or four layers, wear gloves. This clothing is very heavy. [You] have to exert yourself. If the ice is frozen, then you could slip and fall. If I slip and fall, I will have to suffer. I will become a headache to my children.*”—Ranil [59]

#### 3.5.3. Category 1.3 Importance of Clean and Maintained Spaces

IOAs appreciated the cleanliness and maintenance of communal spaces in their neighborhoods. Upon assessment of poorly maintained areas, they felt neglected by those who make decisions about the upkeep of these spaces (e.g., municipality, building owners). Lack of neighborhood cleanliness translated into feeling unsafe in their surroundings and into less motivation to go outside their homes.

“*The roads are dirty and full of cigarette butts and cans. They throw* everything *on the floor. Despite cleaning our front door, they always make it filthy it again…But where can we make a complaint? The municipality never does anything…’ (64-year-old woman)*” [38]

#### 3.5.4. Category 1.4 Availability of Nearby Greenspaces

For IOAs who live a short walking distance from greenspaces, their ability to leverage these accessible areas for recreation and socialization was an important part of their routines. Neighborhood green spaces where people congregate often or where activities for IOAs are held were especially attractive. When aesthetically similar to home country environments, these spaces created a sense of home and motivated neighborhood engagement.

“*Participant 15 (61 years old) said: “I love the beach… I am* from *Puerto Rico, surrounded by water… I love the water. Here I love to go to the water, the water by Downtown.” Participant 15 selected a picture of the beach in Puerto Rico (photo five)…*” [45]

### 3.6. Synthesized Finding 2

Lack of accessibility and safety in transportation caused by discrimination based on language, race, and disability restricted access to valued amenities and social spaces within and beyond the neighborhood which exacerbated social isolation. Transportation access prevents isolation when culturally and linguistically familiar spaces are located outside neighborhood boundaries. This synthesized finding comprised of four categories as below.

#### 3.6.1. Discriminatory Experience in Using Public Transportation

IOAs reported discriminatory attitudes of bus drivers towards them such as a lack of accommodation for those with a disability and racial microaggressions. This resulted in some participants experiencing anxiety and avoiding public transportation.

“*…the mainstream’s view about us makes us feel even lonely. For example, they just talked about the bus. If it is a local Kiwi waiting in that place, the bus would stop. If they see a Chinese person waiting in that place, the driver would not stop the bus. Sometimes when we got on the bus, and ring the bell, they still keep driving and stop at the next bus stop. It took us a long time to walk a long way back. (Chinese man)*” [53]

#### 3.6.2. Having a Car Improves the Quality of Life

IOAs reported having a car enhanced their freedom to travel independently without relying on public transportation or assistance from others and facilitated access to opportunities for socialization. Health challenges preventing IOAs from driving and having to rely on family members for a ride accelerated loss of independence.

“*When we will have no transportation, we will buy a place in the cemetery… Because we will have no life. Without my car, I will not be able to walk to any stores (Russian B)*” [41]

#### 3.6.3. Lack of Access to Public Transportation

IOAs reported the desire for independence when moving around in their neighborhood. Language barriers, lack of bus stops near the home, unreliable bus schedules, and lack of benches and shelters at bus stops limited their ability to use public transportation even when available in the neighborhood. Financial barriers were also a significant concern for IOAs in access to private and public transport, such as buying a bus pass or using taxi services.

“*One Bhutanese older adult said, We get very little money. With that, we have to pay rent. We have to buy food or clothing. We don’t have enough money to spend on the bus. If we get a bus pass, that would be a huge help*” [41]

#### 3.6.4. Valuing Co-Ethnocultural Spaces within Neighborhoods

IOAs reported that a source of well-being and independence was having co-ethnic amenities, services, and programs within their neighborhood that allowed for engagement with co-ethnic neighbors and community members. Co-ethnic spaces in the neighborhood reduced stressors of using transportation to access these spaces outside their neighborhoods.

“*These few blocks, these are my village. Because I know those people. [The] bus is near. And my temple is near. [When] I’m not feeling good I go there. And on Sunday I go and volunteer there… When we bought this house, we thought the gurdwara (temple) should be near—every weekend we should go.*” [58]

### 3.7. Synthesized Finding 3

Racial discrimination, ageism, neighborhood deprivation, and linguistic barriers result in reports of poor social cohesion. Positive neighborhood perceptions were characterized by strong social cohesion via trust, reciprocity, and sense of belonging which was facilitated by having lived in a neighborhood for a long time, knowing one’s neighbors well (irrespective of cultural and linguistic similarities), and the presence of a co-ethnolinguistic community. This synthesized finding comprised of six categories as below.

#### 3.7.1. Safety and Neighborhood Deprivation

Some IOAs also reported that they feel unsafe in their neighborhood. Neighborhood deprivation was reflected in more instances of violence and reports of feeling unsafe by older immigrants which restricted their mobility and social engagement. At times, IOAs were the target of violence and social disruptive behavior and others reported being witnesses with negative impacts on neighborhood perceptions reported in all cases.

“*In the following extract, Tian, a 69-year-old man, described his negative relationship with the neighborhood at the time when eggs were thrown at his daughter’s car. When the first author arrived at Tian’s home for the second interview, Tian was cleaning up his daughter’s car and told the first author: Kids threw chips and eggs at my daughter’s car. It’s frightening. I will remind myself to be careful in the future. For example, lock my doors and windows when I go out, and not to walk closely to a stranger.*” [50]

#### 3.7.2. Racial Discrimination and Ageism

Racial discrimination and ageism were described as overt instances of exclusion or aggression and as covert exclusionary practices, such as not greeting or visiting an older immigrant or not including IOAs in neighborhood decision-making. Importantly, IOAs’ ageist perceptions of younger neighbors also deterred feelings of neighborhood cohesion.

“*Latino kids, kind of acting rowdy, loud.” Later, in response to a question of whether there was anything he did not like about his neighborhood, he added: “What I don’t like is the sense that it’s become a little more dangerous, you know, in terms of reading about assaults, and seeing kids acting out, you know, on the street. You know, fifteen year olds, acting crazy*” [61]

#### 3.7.3. Linguistic Barriers

Some IOAs described how living in ethnic enclaves enabled social cohesion due to cultural and linguistic affinity. Linguistic barriers negatively affecting bridging connections (connecting outside co-ethnic community) in the neighborhood increase isolation. Language barriers coupled with visible markers of differences, such as cultural attire or race, compounded IOAs’ disconnectedness from others in their neighborhood.

“*As expressed by Xin (male, 76 years old, living in Australia for 9 years): “I sometimes see some neighbors in the park. But we don’t communicate much. It’s annoying because I have many words to say but I can’t express myself. I have learnt some simple sentences to communicate with them, but that’s not enough. It’s a pity that we don’t have any in-depth communication.*” [44]

#### 3.7.4. Civic Participation and Reciprocity

While some IOAs may feel isolated in their neighborhoods, others valued their relationships with their neighbors where there was a sense of trust, reciprocity, and safety. Some of these relationships were with neighbors who were from diverse ethnolinguistic backgrounds. IOAs actively created places of belonging via engaging in civic activities and providing support to neighbors.

“*An older Moroccan woman in Brussels, for example, said: I go to the community centre every day. I help with cooking and I’m involved in organising activities so that we can do things together. it’s important to mix with people from different cultural backgrounds. (64-year-old Moroccan woman, 17 years in the neighbourhood, Old-Molenbeek, Brussels)*” [37]

#### 3.7.5. Long-Time Tenure in Neighborhoods

Some IOAs have stayed within familiar neighborhoods in which they have spent many years living here and do not intend to move away because they have developed strong networks of support with neighbors over time, often within co-ethnic communities. Strong relationships with non-co-ethnics were also described when the opportunities were available to develop these relationships.

“*I wouldn’t want to [move away from here] because I don’t want to live far away from all the Turkish people. When I go out here I always meet family and friends on the street; and that gives me a sense of relief. I am already in a foreign country… if I would live somewhere far away from the Turkish community, it would feel as if I’m moving to a foreign country for the second time (69-year-old woman)*” [38]

#### 3.7.6. Changing Neighborhood Composition

Changes in neighborhood composition, particularly the turnover of neighbors over time, can disrupt social cohesion. Similarly, changes in neighborhood composition and neighborhood gentrification can disrupt social cohesion. Neighborhood demographic changes affect the types of amenities in that area such as changes in the types of ethnic shops or services. Many IOAs report disruptions in their sense of belonging and heightened loneliness as the people and aesthetics of neighborhoods change.

“*When I moved in, there were lots of Afro-Caribbean people in the road and lots of White people, and they moved out and Asian people have moved in… The whole area has changed in the last 20 years,” Marjorie described. Millicent highlighted that “all nationality eats different,” and with increasing Eastern European arrivals, food shops in her neighborhood started catering for them: “when you go in looking for something that you’re accustomed to, sometimes they don’t have it…*” [51]

### 3.8. Synthesized Finding 4

Non-neighborhood specific influences, such as pre-migration neighborhood experiences, gendered roles and expectations, and co-residence with family, had strong influences on both the choice of neighborhood to reside in and sense of belonging to the neighborhood. This synthesized finding comprised of three categories.

#### 3.8.1. Home Is Family

Being with family and the quality of familial relationships were key drivers to living in a neighborhood irrespective of feelings of belonging or satisfaction with neighborhood features. Family was the medium through which many IOAs experienced their neighborhood as they relied on family for support with translation, transportation, and access to information about the neighborhood.

“*Ali expressed how much he enjoyed spending leisure time with his grandchildren, playing with them in the backyard and going on daily walks with them around the neighborhood, which kept him “involved seeing how they are doing”. Ali had several grandchildren under the age of five and he explained how these connections filled him with gratitude because he was still physically able to play and build social bonds with them…*” [25]

#### 3.8.2. Gender Shapes Participation

The ways IOAs engage in public neighborhood spaces is tied to gendered roles and expectations. This was seen in two ways. The first was via gendered roles of caregiving and household responsibilities that fell to women and could limit the time, resources, and interest they had in engaging in neighborhood activities such as socializing with neighbors, going for walks, or accessing amenities. “Here I have my grandson,” she continued, “I make my daughter and her family happy; I pray to Buddha. I am peaceful inside [the house] with my small family. Next time [in her next life] maybe I can be free, too.” [49]

The second related to a lack of gender- and age-specific spaces where older migrants felt comfortable to access.

“*I only go shopping in the shop at the corner; I don’t go to [the supermarket] because I have to pass that square then where all the men are (63-year-old woman).*” [38]

#### 3.8.3. Pre-Migration Neighborhood Experiences

Reminiscing on where IOAs have lived in the past illustrates comparisons to their current neighborhoods, which includes considerations of the neighbors they have had across their life course. Neighborhoods post-migration were often described as lacking social cohesion and community-oriented spaces in comparison to home countries. Issues with mobility and weather compounded feelings of isolation in current neighborhoods, whereas neighborhoods in countries of origin were perceived as more walkable, having better weather, and being more familiar.

“*Ana Paula described her life in the Dominican Republic, ‘Life there is never the same as life here… I lived in a… “campo” where you know everyone in the world.” She continued, I came here when I was 35 years old… My daughter be- came a citizen and requested me. After arriving here, well, you already know how life here is. Well here in many ways, one lives a life of ‘having it all.’ And at the same time, with yourself, you have nothing.’ She further explained, ‘Over there, in your own country, you don’t feel loneliness like you feel it here. Over there in the ‘campo’ you open your door in the morning and people say ‘hi, how are you?’ And that’s a different life. Not here. If it’s cold here you have to live in doors…*” [40]

## 4. Discussion

This review focused on IOAs’ perceptions and experiences of living in their neighborhoods. The key contextual factors that were deemed the most important in relation to neighborhood experiences were (a) walkable safe access to greenspaces for socializing nearby, (b) accessible transportation to valued amenities within (and outside of) neighborhoods, (c) social cohesion with neighbors, and (d) influence of pre-migration neighborhood experiences and family.

Conceptual understandings of neighborhoods and their linkage to health in later life have been rapidly growing [7]. The conceptual understandings of AIP have been well-established and defined in relation to the relationships between person, place, and social connections [4,30]. Various theoretical frameworks, such as the capability approach, old-age exclusion, and ecological models, have integrated elements like accessibility, proximity, social connections, and individual characteristics into conceptual models of AIP, especially concerning the neighborhood environment [66,67,68]. A conceptual framework using a capability approach to understand AIP emphasized that place integration and attachment, independence, mobility, and social participation were shaped by the interactions of individual characteristics and environmental factors [67]. A transdisciplinary framework building from the Urban Space Framework elucidates the causal pathways from neighborhood-built environment to older adults’ health by highlighting the mediating role of neighborhood social environments [66]. The Conceptual Framework for Old-Age Exclusion includes six key domains: neighborhood and community, services, amenities and mobility, social relations, material and financial resources, socio-cultural aspects, and civic participation [68]. Findings from this review address the dimensions of these models in particular ways showcasing similarities between IOAs and other older adult groups on ageing in place within neighborhoods. IOAs, like other older adults, require attention and accommodation for physical accessibility and mobility (Categories 1.1, 1.2) [29,67], safety and familiarity (1.1, 1.3) [4,67], and social support (4.1, 4.2) [4,30,68]. Nonetheless, AIP is well known to also vary based on the unique social locations of older adults despite commonalities [67]. This discussion focuses on the unique aspects of IOAs’ neighborhood experiences by emphasizing the inter-relatedness of gender, culture, migration history, and socioeconomic status. Moving beyond established frameworks on AIP, we fill a theoretical gap and focus in on how immigration, discrimination, and cultural differences intersect with AIP experiences [30]. Specifically, we discuss (1) the presence and access to a co-ethnolinguistic community, (2) discrimination across intersections, and (3) incorporation of past neighborhood perceptions across migratory journeys.

### 4.1. Presence and Access to Co-Ethnolingustic Community

In this review, we identified that having co-ethnocultural community nearby reflected in the people and amenities of IOAs’ neighborhoods are fundamental to enhancing personal agency to age well in place. Although this aspect was specifically addressed in Category 2.4, certain principles were also evident in Categories 3.3, 3.4, and 3.6. These categories highlighted social cohesion arising from IOAs’ involvement in co-ethnocultural communities, as well as their varying levels of (dis)engagement or constraints in forming social connections with neighbors outside of these communities. Overall, the psychosocial benefits of being embedded in co-ethnic social networks is well documented in the findings of this review, which is facilitated by living within ethnic enclaves that allow for familiar linguistic and cultural environments. Many IOAs shared the reciprocity and support gained from co-ethnic relationships, or from relationships with those with similar cultures. Gentrification occurred in neighborhoods either in favor of the ethnic composition of the communities congregated over time, or to favor those who are younger or from other ethnocultural backgrounds. This led to social distancing with incoming neighbors due to younger age or different ethnic backgrounds, and disrupted their embeddedness in their neighborhoods.

The literature depicts social connections and building community as fundamental to enacting agency towards AIP [30]. However, for IOAs, social connectedness is not merely a reflection of their preferences but is significantly influenced by their capacity to engage in meaningful and reciprocal social relationships within their neighborhoods. Neighborhood cohesiveness is fundamental for positive neighborhood perceptions [69], and factors such as length of residence may be more directly related to positive neighborhood perceptions over co-ethnic concentration [70]. However, certain groups of IOAs may experience less acculturation stress when living in neighborhoods with higher proportions of co-ethnic members [71]. Our findings did show some ambivalence in regards to the benefits reaped in ethnic enclaves, e.g., [46]. This is echoed in the literature suggesting that increased own-group ethnic density has been associated with a higher level of loneliness among those with good mainstream language fluency but not among those with weaker fluency [72], which brings attention to the interplay and role of bonding and bridging capital amongst IOAs. Further research is required to better understand the role of duration of neighborhood residence, acculturation, and social cohesion amongst co-ethnic in comparison to multi-ethnic neighborhoods for AIP.

### 4.2. Discrimination across Intersections

Participants in the review described their experiences of discrimination as a consequence of their race, ethnicity, physical (dis)ability, age, and/or gender. These experiences bled into their interactions with neighbors (Category 3.2), usage of public transportation (Category 2.1), and built environments and infrastructure (Category 1.1). The findings of this review revealed the lack of support for IOAs with physical ability challenges in being able to access transportation or walk around their neighborhood. Two studies spoke to racism from bus drivers and other patrons using public transportation. On the other hand, IOAs struggled with the social distance from neighbors, resulting in a lack of safety, exclusion, violence, and feelings of fear.

Beyond this review, other studies have exemplified how poor neighborhood perceptions surface from racial discrimination against long-standing ethnic communities and enclaves [73]. Gentrification may leave racialized IOAs feeling culturally displaced despite remaining in the neighborhood [74], especially as gentrification processes will vary by the racial/ethnic composition of the neighborhood [75]. This is of particular concern as older adult and racialized groups are impacted more by gentrification in comparison to White and younger residents, resulting in lower social capital [76], and physical and mental health inequities [77]. As well, the lack of intergenerational cohesion negatively impacts social cohesion for IOAs [78]. As indicated through our findings and echoed in the literature about gentrification, the circumstances that allow for positive neighborhood perceptions and health equity are challenged by systemic levels of social stratification and racialization. Newly arrived IOAs experience aging “out of place” due to experiences of isolation and role loss [12,30]. Rather than characterizing AIP dichotomously (i.e., aging in or out of place), there is a larger question about optimizing the person–environment fit to enable feelings of safety regardless of “place”, which point to larger issues of racialization and discrimination experienced by IOAs. Accessible neighborhoods that can be adapted to the individual to attend to their physical health and desire for agency enhances a sense of belonging [67]. However, our findings have also illuminated that even when older immigrants remain in the same place, changing neighborhoods—whether through shifts in language, cultural composition, or the age of residents—contribute to the dynamic nature of place attachment and neighborhood cohesion across the life course [67,79]. This requires attention to the way neighborhoods are designed to attend to the needs of all residents and enhance social cohesion regardless of their age, health status, life stage, race, or other circumstances.

### 4.3. Influence of Pre-Migration Neighborhood Experiences and Family

Despite the primary focus on local neighborhood perceptions, IOAs often drew comparisons to their neighborhood and community experiences before and during migration. As exemplified in Category 4.3, some of these comparisons drew on the past to conceptualize how they were now aging out of place due to isolation, less walkable built environments, and lack of culturally familiar recreational activities. Other comparisons evoked markers of their country of origin which were important for AIP in their neighborhoods. This further strengthened emotional ties to place throughout their lives, influenced by past cultures and family support as identified in Category 4.1. Additionally, as exemplified in Category 1.4, certain neighborhood features, like natural environments, supported AIP and health promotion across time and space.

IOAs may not be living exclusively in post-migratory environments and may experience dual belonging in both post-migratory contexts and their countries of origin [80]. A theoretical examination of AIP using socioemotional selectivity theory has illuminated potential reasons behind decisions and motivations to remain in one location rather than relocating [79]. This decision-making process may be more complex for IOAs and their families, who must consider how they will AIP in relation to their country of origin, their current post-migratory context, and other possible locations that may better meet their needs. However, this review was focused on the local post-migratory neighborhood context, and did not focus on the concept of home-making beyond that. The existing body of literature on AIP could also benefit from a more comprehensive examination of transnational belonging [81], highlighting the need for studies that can broaden our spatial understanding of neighborhoods to encompass various transnational settings. This could involve research designs that compare IOAs and other older adults living in their country of origin, and employ longitudinal approaches to capture neighborhood experiences over time. Furthermore, temporary migrants were not the focus in this review, and thus further investigation to understand how immigrant status and migration history shape place attachment and AIP is warranted. This is critical to further understandings beyond aging in (or out) of place, and incorporate the reality of individual agency in the context of global mobility and access to multiple environments that may enable support for IOAs aging globally. This may add to the literature that has traditionally situated AIP in the local home/neighborhood/community, and situated an ideological focus on “staying in one place” which may not apply to individuals subjected to patterns of migration, travel, and global connectivity. Paying attention to how age-friendly neighborhoods are constructed and enforced globally beyond Anglo-dominant societies may provide further insights that enhance resources so that IOAs may exercise agency over sustaining their livelihoods as they age.

This review findings shed light on the significance of family support in influencing IOAs’ engagement in neighborhood activities and their sense of belonging. The presence or absence of such support networks plays a crucial role in determining their level of engagement and feelings of connectedness within their neighborhoods. However, this aspect warrants further investigation. IOAs who have poor family supports are at a higher risk for experiencing loneliness and isolation, and may face increased reliance on services and encounter barriers in accessing care [82,83,84]. This reliance on family support networks is not only cultural but also structural, as families often step in to address gaps in culturally and linguistically relevant service provision [84,85]. Within neighborhood contexts, families may serve as substitutes for the absence of supports for AIP in this population, which raises issues related to unpaid care work and equity [4]. Conversely, living in co-ethnolinguistic communities in the absence of family supports might be a protective factor but evidence is inconclusive based on this review. Research suggests that while both family and co-ethnic community networks are vital for fostering a sense of belonging by eliminating language barriers, reciprocal relationships within these networks emerge as more crucial for fostering positive feelings of belonging and connection among IOAs [86]. Therefore, understanding the intricate dynamics of these networks and supports is essential for promoting the well-being of IOAs and facilitating successful AIP.

### 4.4. Strengths and Limitations

Strengths of this review include the use of a robust systematic review methodology which centers on meta-aggregation, an approach that centers the lived experiences of participants within the secondary analysis. As well, the included studies covered a range of populations across contexts, including diverse countries of origin for participants, urban/rural settings, and the inclusion of ethnic enclaves and non-ethnic enclaves, which strengthens the transferability of the findings in this review. Finally, the authors are experts in aging, systematic reviews, qualitative research, and migration, with a combination of lived experiences from within and outside immigrant and racialized communities. This inherently strengthened our analysis and allowed for reflexivity across these diverse experiences and areas of expertise.

One limitation of this review concerns the reporting of qualitative rigor of the included studies. Despite efforts to be more inclusive by marking studies that did not explicitly state their philosophical perspective in methodology as “Yes” so long as there was a theoretical framework guiding their study, there still was a lack of congruity between philosophical perspectives and research methodology. Studies were also not consistent with reporting researcher positionality. Future research could benefit from explicit statements of researcher positionality, methodology, and philosophical approach. Additionally, this review lacks representation of other populations such as those aged over 85 years, those who are temporary migrants, and LGBTQS2+ immigrant older adults. Furthermore, ambiguity in conceptualizing “neighborhoods” [26], may have resulted in the exclusion of studies that did not clearly reference “neighborhoods” but were referencing geographical areas in proximity to where immigrants lived. While we sought to include studies that addressed community in proximity to where immigrants lived, this was not always clear from the study objectives or findings. Although attempts were made to capture neighborhood-related experiences, some studies approached neighborhoods as part of broader research objectives, possibly diluting their focus on neighborhood experiences.

## 5. Conclusions

The findings of this review contribute to the ongoing discussion about the diverse nature of AIP, highlighting that while there are commonalities among IOAs, individual experiences vary based on factors influencing their sense of well-being and belonging. Consequently, adapting neighborhood environments to suit each individual is challenging due to differing needs and desires. In addressing AIP, it is crucial to prioritize flexible, community-driven approaches, backed by sustainable funding for essential transportation and venues, both within and beyond their immediate neighborhoods. By co-creating neighborhoods that deter discrimination and microaggressions, while empowering IOAs who have felt marginalized, they may engage in civic opportunities and have their voices heard. The idea of “neighborhood” is shaped by social constructs and connections to people and places. Both fostering close-knit relationships and building bridges to new connections are vital for providing consistent social support aligned with IOAs’ expectations.

## Figures and Tables

**Figure 1 ijerph-21-00904-f001:**
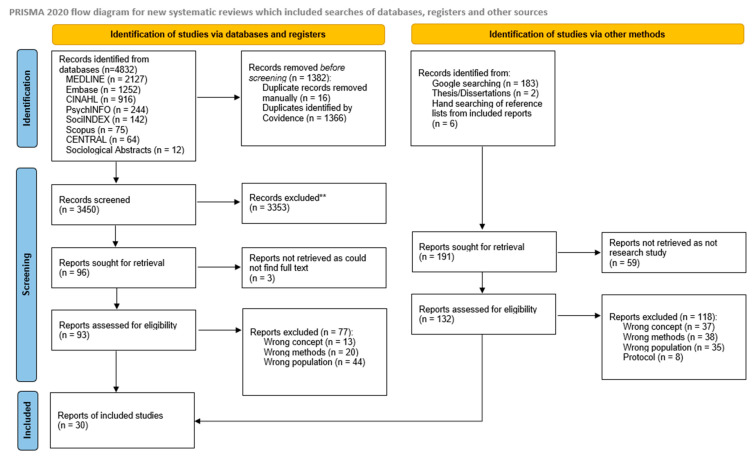
PRISMA Diagram: Search results and study selection and inclusion process [65].

**Table 1 ijerph-21-00904-t001:** Critical Appraisal Results of Included Studies.

JBI Critical Appraisal Checklist for Qualitative Research Item:	Included Studies																													
	Becares 2013 [34]	Brotman 2017 [35]	Buffel 2011 [36]	Buffel 2013 [37]	Buffel 2017 [38]	Chen 2022 [39]	Curtin 2017 [40]	Dabelko-Schoeny 2021 [41]	Dorkenoo 2021 [42]	Fang 2016 [43]	Gao 2020 [44]	Hawkins 2022 [45]	Herman 2021 [46]	Hsu 2014 [47]	Jagroep 2023 [48]	Lewis 2009 [49]	Li 2014 [50]	Lorinc 2022 [51]	Luo 2016 [52]	Morgan 2021 [53]	Nasir 2022 [25]	Parekh 2018 [54]	Rua 2017 [55]	Ryan 2021 [56]	Schuster 2019 [57]	Tong 2020 [58]	Wijekoon 2018 [59]	Xu 2023 [60]	Yen 2012 [61]	Zhan 2017 [62]
Q1	Y	Y	Y	Y	U	U	Y	Y	Y	Y	Y	Y	Y	U	Y	Y	N	U	N	U	Y	Y	Y	Y	Y	Y	Y	Y	Y	Y
Q2	Y	Y	Y	Y	Y	Y	Y	Y	Y	Y	Y	Y	Y	Y	Y	Y	U	Y	Y	Y	Y	Y	Y	Y	Y	Y	Y	Y	Y	Y
Q3	Y	Y	Y	Y	Y	Y	Y	Y	Y	Y	Y	Y	Y	Y	Y	Y	Y	Y	Y	Y	Y	Y	Y	Y	Y	Y	Y	Y	U	N
Q4	Y	Y	Y	Y	Y	Y	Y	Y	Y	Y	Y	Y	Y	Y	Y	Y	Y	Y	Y	Y	Y	Y	Y	Y	Y	Y	Y	Y	Y	Y
Q5	Y	Y	Y	Y	Y	Y	Y	Y	Y	Y	Y	Y	Y	Y	Y	Y	Y	Y	Y	Y	Y	Y	Y	Y	Y	Y	Y	Y	Y	Y
Q6	U	U	Y	N	Y	Y	Y	Y	N	Y	Y	Y	Y	U	Y	Y	Y	U	Y	U	Y	U	Y	U	Y	U	Y	Y	N	Y
Q7	U	U	Y	U	Y	U	Y	Y	U	Y	Y	Y	Y	Y	Y	Y	U	Y	Y	U	Y	U	Y	U	Y	N	Y	Y	N	N
Q8	Y	Y	Y	Y	Y	Y	Y	Y	Y	N	Y	Y	Y	Y	Y	Y	Y	Y	Y	Y	Y	Y	Y	Y	Y	Y	Y	Y	Y	Y
Q9	Y	U	N	U	U	Y	Y	Y	Y	Y	Y	Y	N	Y	Y	N	U	Y	Y	Y	Y	Y	N	Y	Y	Y	Y	Y	Y	Y
Q10	U	U	U	Y	Y	Y	Y	Y	Y	U	Y	Y	Y	Y	Y	Y	Y	Y	Y	Y	Y	Y	Y	Y	Y	Y	Y	Y	Y	Y
Total “Yes” (out of 10)	7	6	8	7	8	8	10	10	8	8	10	10	9	8	10	9	6	8	9	7	10	8	9	8	10	8	10	10	7	8

Note: Y = Yes; N = No; U = Unclear. Dependability was assessed using Q2, Q3, Q4, Q6, and Q7.

**Table 2 ijerph-21-00904-t002:** Credibility Assessment.

Level of Credibility	Criterion
Unequivocal	Illustrations may be quotations or rich thick descriptions through observational data and/or contextual data Must be clear to the reviewer that the author’s statement was closely aligned with the illustration
Credible	Illustrations may be quotations or rich thick descriptions through observational data and/or contextual data Author’s statements appeared to be a conceptual leap from the illustration
Unsupported	Author statements do not have an accompanying illustration

**Table 3 ijerph-21-00904-t003:** Study Characteristics.

First Author and Year	Methodology/Method/Analysis	Phenomena of Interest	Setting	Participants
Becares 2013 [34]	Mixed methods; face to face interviews Content Analysis: Framework	The association between ethnic density, social capital, and health, in order to establish whether social capital mediates the association between ethnic density and health among ethnic minority groups in England.	England	18 Jamaican Caribbean and 15 Gujarati Indian Hindu older adults aged between 65 and 74 years.
Brotman 2017 [35]	Narrative photovoice; in-depth narrative interviews and photographs Intersectionality, critical life course, and photovoice framework used for analysis	Lived experiences of immigrant seniors. The impact of immigration on aging within the context of life histories. To understand the intersections of identity, social location, and structural discrimination across the lifespan. To explore the ways in which structural discrimination across the life-course shapes interactions with family, community, and formal services.	Canada: Quebec and British Columbia	19 older adult immigrants.
Buffel 2011 [36]	Semi-structured interviews Thematic Analysis	To explore experiences of “place” among older migrants living in deprived urban neighbourhoods.	Belgium (Brussels) and England (Manchester)	20 Moroccan and 23 Turkish participants living in Brussels, Belgium, aged between 60 and 73. 19 Somali participants in Liverpool, England, and 20 Pakistani participants in Manchester, England.
Buffel 2013 [37]	Semi-structured interviews Thematic Analysis	Conceptual and empirical aspects of the social exclusion debate, exploring links with issues of place and space in urban settings in two contrasting European nations.	Belgium (Brussels) and England (Manchester)	124 British seniors with Pakistani, Somali, or Black Carribean origin, 102 Belgium seniors with Turkish or Moroccan origin aged 60 and over.
Buffel 2017 [38]	Interviews Combination of thematic and content analysis	The ways in which ageing migrants experience the notion of ‘home’, both as a location and a set of relationships that contribute to feelings of belonging and identity.	Belgium; inner-city districts of Brussels	34 first-generation Turkish labor migrants living in two neighboring districts in Brussels, 18 women and 16 men aged between 60 and 78.
Chen 2022 [39]	Semi-structured, face-to-face interviews Constructive grounded theory analysis	To identify whether Chinatowns are a place for Chinese immigrants to age and explore their experience of aging in Chinatowns.	USA; New York City’s Chinatowns	22 older adults (14 females) aged 60 years or older, originally migrated from China, previously lived in or currently live in one of three Chinese-immigrant cluster areas. Eight participants were over 75.
Curtin 2017 [40]	Inductive qualitative descriptive design; semi-structured interviews Inductive analysis	Ageing out of place and the meaning of home for a group of older persons of Hispanic ethnicity.	USA; New England	17 Hispanic older persons, 11 women and 6 men aged between 65 and 83, average age of 71 years.
Dabelko-Schoeny 2021 [41]	Concurrent focus group discussions Rapid and Rigorous Qualitative Data Analysis (RADaR) technique and thematic analysis with an interactive team approach	Comprehensive understanding of the factors affecting transportation among diverse older adults.	USA; Central Ohio	70 older adult volunteers represented culturally diverse immigrant and refugee communities, 40% men and 60% women.
Dorkenoo 2021 [42]	Mixed method, sequential, equal-status; group interview, surveys GIS qualitative approach and thematic mapping for qualitative data, spatial analysis for quantitative data	Investigate the formal and informal social supports of Arabic-, Mandarin- and Spanish-speaking older immigrants in the City of Toronto, specifically what their experiences were as they age in place.	Canada; Ontario—Greater Toronto Area (GTA), Ottawa, Waterloo and London.	95 participants aged 55 and over, speak Arabic, Mandarin, or Spanish as their native language.
Fang 2016 [43]	Community-based participatory research; experiential group walks and participatory mapping exercises with visual (photograph) data Thematic analysis	Access experiences of place, identify facilitators and barriers to accessing the built environment and co-create place-based solutions among older people and service providers in a new affordable housing development in Western Canada.	Canada (Western)	54 participants (N = 38 English and Mandarin-speaking older persons with diverse cultural backgrounds over the age of 60; N = 16 local service providers).
Gao 2020 [44]	Mixed-method case study; travel diaries, mapping, and interviews Comparative method qualitative data analysis	Understanding green spaces and how they influence the well-being of older Chinese immigrants, within an Australian multicultural context through a lens situated in Chinese values and beliefs. Provide insights into why and how designing green spaces can better maintain the well-being of elderly immigrants in Australia.	Australia; City of Gold Coast; public parks, churches, and Chinese community centers	30 Chinese immigrants aged 55 and above.
Hawkins 2022 [45]	Community-engaged participatory narrative inquiry photovoice; photos and in-depth interviews Narrative inquiry and Barone and Eisner’s criteria for arts-based work	To explore the health influence and experience of older adult Russian- and Spanish-speaking English in southeastern Wisconsin, United States.	USA; southeastern Wisonsin	23 older adult female Russian- and Spanish-speaking immigrants between the ages of 60 and 98.
Herman 2021 [46]	Semi-structured, face-to-face interviews Inductive and iterative analysis	To examine the experiences of members of Saskatoon’s Chinese–Canadian older-adult community in terms of their realities of aging and access to important geriatric resources.	Canada; Saskatoon	20 participants aged 55 or older, self-identify as Chinese–Canadian, no limitation on how long they had resided in Canada.
Hsu 2014 [47]	Interviews, narratives No description of analysis	Claiming home through exercising agency and working out paradoxes concerning their living conditions, familial relations, and subjectivities.	Canada; Montreal’s Chinatown	25 interviews with Chinese female seniors residing in Montreal’s Chinatown. Average age of 79 years. Their average age was 50 years when they moved to Canada, and the average time spent here was 25.58 years.
Jagroep 2023 [48]	Semi-structured interviews Thematic analysis	Explore how older Surinamese adults experienced their neighbourhood age-friendliness in general and during the COVID-19 pandemic.	Netherlands; Rotterdam or the Hague	17 participants who were aged 70 or older, had a Surinamese migration background, the ability and willingness to answer questions in Dutch, and were community-dwelling (independently living).
Lewis 2009 [49]	Ethnography; semi-structured and unstructured interviews, participant observations Ethnographic analysis, open and focused coding	Experiences and expectations of elderly Cambodian refugees who are aging out of place, far from the familiar cultural, social, and political landscape in which they lived and in which they developed an understanding of what it meant to be “old”.	USA; rural community in coastal Alabama	Elderly Cambodian participants from 125 families, aged 55 years and older.
Li 2014 [50]	Semi-structured interviews Narrative thematic analysis	Investigate older Chinese migrants’ experiences of developing a sense of community (SOC) in their local communities. To better understand how everyday activities, processes, and practices foster multiple SOCs, the second section explores their SOC within the context of transnationalism.	New Zealand	32 older Chinese migrants who ranged in age from 62 to 77 years; 18 female and 14 male.
Lorinc 2022 [51]	Longitudinal, qualitatively driven multi-method study; in-depth interviews and follow-up walking interview Inductive thematic analysis	Investigate experiences of aging for older migrants in England, focusing on their well-being, care needs, and support.	England: London and Yorkshire	45 older migrants from the Caribbean, Ireland, and Poland, the majority were older than 80; 9 were selected for follow-up interviews.
Luo 2016 [52]	Focus group interviews Six-step thematic analysis	To explore the perceptions and experiences of older Chinese immigrants regarding their current life, social capital level, and residential environment, as well as their expectations of social capital and residential environment. The study also aimed to identify implications for policies and practices to improve the social capital of older Chinese immigrants through supportive living environments.	Canada	43 Chinese immigrants born outside of Canada, the majority between 75 and 84 years. The participants had been living in Canada for an average of 15.8 years, with a range from 1 to 43 years. The majority of the participants were originally from mainland China, with a small number from Vietnam and Hong Kong.
Morgan 2021 [53]	Mixed methods, intial qualitative phase; semi-structured individual and group interviews Thematic and narrative analysis	Older people’s protective factors that enable or foster social connectedness, and factors that prevent or operate as barriers to social connectedness.	New Zealand; Aotearoa	Diverse group of Pacific, Māori, Asian, and New Zealand European older adults. 44 participants took part in individual in-depth interviews and 32 participants took part across three group discussions.
Nasir 2022 [25]	Constructivist narrative inquiry; narrative interview and follow-up sessions Thematic analysis	To understand the social relationships of aging Muslim Lebanese immigrants living in Canada by exploring their lives in their ethnic and wider communities.	Canada; London, Ontario	4 participants who were English or Arabic speaking, Muslim immigrants from Lebanon, 60 years of age and over, and immigrated to Canada in early adulthood.
Parekh 2018 [54]	Community-based participatory research study; individual and focus group interviews Interpretive qualitative framework analysis	To explore the role of social capital (e.g., social support through indirect ties) and social cohesion (e.g., interdependent support among neighbors) to unravel pathways for building age-friendly communities.	USA	Older adults aged 55 and over, with African-American, Hispanic, or Vietnamese descent; 15 participants for individual interviews and 45 participants for focus group interviews.
Rúa 2017 [55]	Narrative inquiry and ethnography; participant observation, collection of life histories, and semi-structured interviews	To examine how urban revitalization processes in Chicago are impacting the experiences and well-being of Puerto Rican elderly individuals who have been displaced from their communities.	USA; Chicago	25 Latina and Latino older adults living in three different subsidized housing complexes on Chicago’s Near Northwest Side.
Ryan 2021 [56]	Interview and walking interviews Thematic analysis	To understand how older migrants, especially in advanced old age, navigated ageing and care in place.	England; London and Yorkshire	45 older migrants from the Caribbean, Ireland, and Poland, the majority were older than 80; 9 were selected for follow-up interviews.
Schuster 2019 [57]	Narrative inquiry; face-to-face interviews in three sessions Inductive narrative analysis	Identify the roles recent older Canadian immigrants play within their families and communities, the challenges, facilitators, and pressures they encounter in doing so, and the benefits or drawbacks they experience regarding contributing to their surroundings.	Canada	4 older immigrant participants who have lived in Canada for at least 5 years and moved to live with family in Canada.
Tong 2020 [58]	Ethnography; in-depth semi-structured interviews	To characterize the PA habits of multilingual and non-English-speaking FBOAs who reside in South Vancouver, British Columbia, Canada.	Canada; Vancouver	18 Chinese or South Asian older adults.
Wijekoon 2018 [59]	An interpretive paradigm and hermeneutic phenomenology; phenomenological interviews and photo-elicitation interviews Interpretive phenomenological analysis (IPA)	Understand how late-life immigrants Relate to, and connect and engage with, places through aging processes, and the essentiality of daily occupations within such engagement.	Canada; Ontario, Greater Toronto Area (GTA)	10 participants aged between 72 and 82. All participants immigrated from Sri Lanka to Canada under the Parent and Grandparent (PGP) Sponsorship Program between 2007 and 2013.
Xu 2023 [60]	Face-to-face interviews Thematic analysis	To explore social exclusion and its risk factors among older Chinese adults in greater Los Angeles.	USA; Los Angeles	24 Chinese Americans aged 65+.
Yen 2012 [61]	Face-to-face interviews Thematic analysis	Identify the types of resources that people use in their residential settings to maintain or improve their overall well-being.	USA; San Francisco Bay Area	38 participants aged 65 and over who self-identified as White, African American, Latino, or Asian American
Zhan 2017 [62]	Surveys and interviews	Examine Chinese immigrant elders’ report of their sense of home and life satisfaction.	USA; Atlanta	107 Chinese American participants completed surveys and 21 participants completed interviews. Participants ranged from 59 to 93 years.

**Table 4 ijerph-21-00904-t004:** Meta-Aggregation Categories and Synthesized Findings.

Synthesized Finding:	Categories
1. Walkability can be improved by enhancing safety features of neighborhoods to address physical disability limitations that are exacerbated by winter weather conditions. Preferences about living in areas with warmer climate, clean spaces, and accessible greenspaces that allow for socialization were important factors for IOAs to feel comfortable in their neighborhood.	1.1 Physical accessibility and walkability
	1.2 Winter weather challenges
	1.3 Importance of clean and maintained spaces
	1.4 Availability of nearby green spaces
2. Lack of accessibility and safety in transportation caused by discrimination based on language, race, and disability restricted access to valued amenities and social spaces within and beyond the neighborhood which exacerbated social isolation. Transportation is most critical to prevent isolation when culturally and linguistically familiar spaces are located outside neighborhood boundaries.	2.1 Discriminatory experience in using public transportation
	2.2 Having a car improves the quality of life
	2.3 Lack of access to public transportation
	2.4 Valuing co-ethnocultural spaces within neighborhoods
3. Racial discrimination, ageism, neighborhood deprivation, and linguistic barriers result in reports of poor social cohesion. Positive neighborhood perceptions were characterized by strong social cohesion via trust, reciprocity, and sense of belonging which was facilitated by having lived in a neighborhood for a long time, knowing one’s neighbors well (irrespective of cultural and linguistic similarities), and the presence of co-ethnolinguistic community.	3.1 Safety and neighborhood deprivation
	3.2 Racial discrimination and ageism
	3.3 Linguistic barriers
	3.4 Civic participation and reciprocity
	3.5 Long-time tenure in neighborhoods
	3.6 Changing Neighborhood Composition
4. Non-neighborhood-specific influences—premigration neighborhood experiences, gendered roles and expectations, and co-residence with family—had strong influences on both the choice of neighborhood to reside in and sense of belonging to the neighbourhood.	4.1 Home is family
	4.2 Gender shapes participation
	4.3 Pre-migration neighborhood experiences

Appendix A: Full Meta-Aggregation.

**Table 5 ijerph-21-00904-t005:** ConQual Summary of Findings.

Population: Immigrant Older Adults (IOAs)					
Phenomena of Interest: Experiences and Perceptions of Neighborhoods					
Context: Any Country; Geographical Space in Proximity to where Older Adults Live					
Synthesized Finding	Type of Research	Dependability	Credibility	ConQual Score	Comments
Synthesized finding 1	Qualitative	High (No downgrading)	Moderate (Downgrade one level)	Moderate	Dependability: Majority of studies (11/13) scored 4 and 5 for the questions relating to appropriateness of the conduct of the research. Credibility: Downgraded one level due to mix of unequivocal (U) and credible (C) findings. U = 28, C = 4
Synthesized finding 2	Qualitative	High (No downgrading)	Moderate (Downgrade one level)	Moderate	Dependability: Majority of studies (15/25) scored 4 and 5 for the questions relating to appropriateness of the conduct of the research. Credibility: Downgraded one level due to mix of unequivocal (U) and credible (C) findings. U = 70, C = 9
Synthesized finding 3	Qualitative	High (No downgrading)	Moderate (Downgrade one level)	Moderate	Dependability: Majority of studies (17/26) scored 4 and 5 for the questions relating to appropriateness of the conduct of the research. Credibility: Downgraded one level due to mix of unequivocal (U) and credible (C) findings. U = 95, C = 16
Synthesized finding 4	Qualitative	High (No downgrading)	Moderate (Downgrade one level)	Moderate	Dependability: Majority of studies (11/15) scored 4 and 5 for the questions relating to appropriateness of the conduct of the research. Credibility: Downgraded one level due to mix of unequivocal (U) and credible (C) findings. U = 41, C = 5

U: unequivocal; C: credible.

## Data Availability

The authors confirm that the data supporting the findings of this study are available within the article and its Appendix A.

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
