# Peer review of "Immigrant Older Adults’ Experiences of Aging in Place and Their Neighborhoods: A Qualitative Systematic Review"

_ijerph, 2024, doi:10.3390/ijerph21070904_

Round 1
Reviewer 1 Report
Comments and Suggestions for Authors
Dear Respectable Authors
Thank you for considering a significant area of research related to immigrants and aging. You conducted a qualitative systematic review to investigate the immigrant older adults' perceptions and experiences of the neighborhoods in which they live. Your results are of interest but your manuscript needs some revisions as follows. I hope my comments will better the quality of your reporting.
- Abstract, please add the date of search and search period.
- Abstract, please add the eligibility criteria.
- Abstract, please add the way you appraise the results. Which checklist do you use?
- Abstract, please add some results regarding the summary characteristics of the included studies.
- Introduction, paragraph 4, references 17-21 are redundant. There is one statement and five citations. It is not rational.
- Introduction, please add the aim of the study at the end of this section. Other items mentioned after the research question are related to the study method and should be removed from this section.
- Methods, please replace eligibility criteria before the search strategy. Please follow the PRISMA statement.
- Please remove Table 1 from the methods section and replace it in the results section.
- Quality appraisal, you stated that score studies in a 10-scale format but in Table 1 the overall scores differ. It is not transparent. Maybe it is better to remove the last column.
- Table 1, please use the first author and year instead of the number of citations in Table 1 and all tables. Also, you can remove the full question and only mention the number of questions in Table 1, first column.
- Please remove Table 2 from the methods. This Table is related to the results. Also, use the first author name and year for column one of Table 2.
- Please remove all tables from the methods. All Tables are related to results.
- Please replace the Figure 1 before the caption. Also, your figure is not complete. Please look at the PRISMA flow diagram and edit this Figure. Some lines were missed.
- Please add the quality appraisal results after the characteristics of the included studies. It is not based on the PRISMA.
- The other results and discussion are well-written.
Cheers
Author Response
Thank you to the editorial team and reviewer for the valuable feedback. We have responded to the comments and have used highlights to indicate where in the manuscript these comments have been addressed. We are happy to provide further clarity as needed.
Comments 1: Abstract, please add the date of search and search period.
Response 1: Thank you for your comment, we have made this change accordingly in the abstract.
Comments 2: Abstract, please add the eligibility criteria.
Response 2: Thank you for your comment, we have made this change accordingly in the abstract.
Comments 3: Abstract, please add the way you appraise the results. Which checklist do you use?
Response 3: Thank you for your comment, we have made this change accordingly in the abstract.
Comments 4: Abstract, please add some results regarding the summary characteristics of the included studies.
Response 4: Thank you for your comment, we have made this change accordingly in the abstract.
Comments 5: Introduction, paragraph 4, references 17-21 are redundant. There is one statement and five citations. It is not rational.
Response 5: Thank you for your comment, we have made this change accordingly in the introduction.
Comments 6: Introduction, please add the aim of the study at the end of this section. Other items mentioned after the research question are related to the study method and should be removed from this section.
Response 6: We agree, the have rearranged the information in the introduction to enhance readability.
Comments 7: Methods, please replace eligibility criteria before the search strategy. Please follow the PRISMA statement.
Response 7: Thank you for your comment, we have made this change accordingly.
Comments 8: Please remove Table 1 from the methods section and replace it in the results section.
Response 8: Thank you for your comment, we have made this change accordingly.
Comments 9: Quality appraisal, you stated that score studies in a 10-scale format but in Table 1 the overall scores differ. It is not transparent. Maybe it is better to remove the last column
Response 9: Thank you for your comment. We have changed Table 1 to include more information. We have also rectified the text in the methods to explain that studies were scored using the 10 questions and received a point for each “Yes” response. As per your suggestion, we have opted to remove the last column of the table to simplify what readers are looking at.
Comments 10: Table 1, please use the first author and year instead of the number of citations in Table 1 and all tables. Also, you can remove the full question and only mention the number of questions in Table 1, first column
Response 10: Thank you for your comment, we have made this change accordingly
Comments 11: Please remove Table 2 from the methods. This Table is related to the results. Also, use the first author name and year for column one of Table 2.
Response 11: Thank you for your comment, we have made this change accordingly.
Comments 12: Please remove all tables from the methods. All Tables are related to results.
Response 12: Thank you for your comment, we have made this change accordingly.
Comments 13: Please replace the Figure 1 before the caption. Also, your figure is not complete. Please look at the PRISMA flow diagram and edit this Figure. Some lines were missed.
Response to C13: Thank you for your comment, we have uploaded a revised version of the figure.
Comments 14: Please add the quality appraisal results after the characteristics of the included studies. It is not based on the PRISMA.
Response 14: Thank you for your comment, we have made this change accordingly.
Reviewer 2 Report
Comments and Suggestions for Authors
The article strenghts and limitation section comprehensively refers to the potential issues with the study. I'd personally would like to add that the topic would merit different research approach that doesn't rely on secondary sources via metaanalysis, but rather is conducted as a proper population study.
Author Response
Thank you to the editorial team and reviewer for the valuable feedback. We have responded to the comments and have used highlights to indicate where in the manuscript these comments have been addressed. We are happy to provide further clarity as needed.
Comments 1: The article strengths and limitation section comprehensively refers to the potential issues with the study. I'd personally would like to add that the topic would merit different research approach that doesn't rely on secondary sources via meta-analysis, but rather is conducted as a proper population study
Response 1: Thank you for your comment. Building on the work done in this review, our team has initiated original community-based participatory research to enhance the literature we analyzed. We hope our primary research will address some of the gaps and limitations identified in the existing studies. We look forward to continuing our research on this important topic.
Reviewer 3 Report
Comments and Suggestions for Authors
Thank you for your important work-- this review will advance science and addresses an urgent oversight in AIP literature. I enjoyed reading it, and thought your procedures and findings were well-described. The manuscript can be improved in a few key ways:
-
A brief operationalization of “aging in place” is needed before going too far.
-
The thesis of the first paragraph on p 2 is that IOA’s have many different experiences than domestic-born OA. Any statement you make about aging in place should be theoretical or framed as hypothesis given the aims of your review and the statements about lack of synthesis on this subject
-
The sequence/flow of the introduction may benefit from reordering of paragraphs once these other suggestions are taken
-
The evaluation of the quality of research is less important than the findings and discussion, given that quality wasn't part of your initial research question. While I understand it is part of review procedures and is relevant, it may be worth saving your word count in your revision process for more explication of results/discussion.
-
This is likely an editorial decision, but it may be easier to comprehend if the inclusion section was condensed into a singular section. What is PICo– cite.
-
Justification is needed for your timelines. Why not select within the last ten years?
-
Did you use specific procedures for screening and reconciliation? How does the JMI guide that?
-
Table 3 is actually the key that I was looking for to understand Table 1-- either place within this first table or reorder.
-
I noticed that the included articles do not seem to cover research using the term “expat”/ “expatriate”. Whether or not this is true, it could be worth considering and/or explaining in your discussion of inclusion or exclusion, your limitations section, etc. I first noticed this when reading the characteristics of the included studies, as study settings were primarily Anglo– should speak to racialization and global hegemony.
-
The results section of this manuscript is compelling and well-organized, but the discussion doesn’t do enough to bridge your findings into AIP frameworks. AIP is extremely well-established as a concept, and I expect the discussion to add to the AIP conceptual model. The following literature may assist: Forsyth & Molinsky, 2021; Rosenwohl-Mack et al., 2020; Ahn, Kang, & Kwon, 2020; Bigonnesse & Chaudhury, 2020; Bigonnesse & Chaudhury, 2021; Golant, 2008, 2009, 2011, 2017, 2018; Canham et al., 2022; Zang, Loo, & Wang, 2022; Weil & Smith, 2016; Lehning et al., 2017; Versey, 2021
Author Response
Thank you to the editorial team and reviewer for the valuable feedback. We have responded to the comments and have used highlights to indicate where in the manuscript these comments have been addressed. We are happy to provide further clarity as needed.
Comments 1: A brief operationalization of “aging in place” is needed before going too far
Response 1: Thank you for your suggestion. We agree and have now defined “aging in place” clearly and early on in the paper.
Comments 2: The thesis of the first paragraph on p 2 is that IOA’s have many different experiences than domestic-born OA. Any statement you make about aging in place should be theoretical or framed as hypothesis given the aims of your review and the statements about lack of synthesis on this subject
Response 2: We agree with this comment, and have incorporated more detail in the aforementioned paragraph to ensure we are presenting theoretical propositions and hypotheses.
Comments 3: The sequence/flow of the introduction may benefit from reordering of paragraphs once these other suggestions are taken
Response 3: We agree that the readability of the introduction would benefit from restructuring. We have taken the liberty to enhance the flow which required in-depth edits. We did not add new ideas or writing, thus there are no highlights apparent.
Comments 4: The evaluation of the quality of research is less important than the findings and discussion, given that quality wasn't part of your initial research question. While I understand it is part of review procedures and is relevant, it may be worth saving your word count in your revision process for more explication of results/discussion.
Response 4: We agree with this comment and have shortened our assessment of methodological quality accordingly.
Comments 5: This is likely an editorial decision, but it may be easier to comprehend if the inclusion section was condensed into a singular section. What is PICo– cite.
Response 5: We agree with this comment and have removed the headings and PICo and discussed the inclusion criteria more broadly.
Comments 6: Justification is needed for your timelines. Why not select within the last ten years?
Response 6: Thank you for your important comment. We decided to not select within the last ten years to ensure we are being as inclusive as possible. There is not a lot of literature in this area so being inclusive versus restrictive would maximize understanding of the issue.
Comments 7: Did you use specific procedures for screening and reconciliation? How does the JMI guide that?
Response 7: Thank you for pointing this out. A JBI review requires a minimum of two reviewers to conduct a systematic review to adequately complete the work to the standards dictated in their Manual. Screening and data extraction was completed by two researchers and reconciliation occurred in team meetings where consensus was sought for discrepancies. As well, the review was led by senior researchers experienced in systematic reviews.
Comments 8: Table 3 is actually the key that I was looking for to understand Table 1-- either place within this first table or reorder.
Response 8: Thank you for your suggestion. After some consideration, we opted to move all the tables containing results to the results section. We have left Table 3 (it is the new Table 1) in the methods section and it assists with understanding the supplemental files (for individual studies) and the ConQual table (Table 4). We hope that the reordering of tables helps readers understand the information we are presenting. We are open to rearranging the tables further as needed.
Comments 9: I noticed that the included articles do not seem to cover research using the term “expat”/ “expatriate”. Whether or not this is true, it could be worth considering and/or explaining in your discussion of inclusion or exclusion, your limitations section, etc. I first noticed this when reading the characteristics of the included studies, as study settings were primarily Anglo– should speak to racialization and global hegemony.
Response 9: Thank you for bringing forward this important point. We recognize that we did not include “expat”/ “expatriate” in our search strategy. However, our understanding is that an expat is someone who does not intend to stay in the new country of residence while an immigrant is permanent residence in the new country. The term migrant encompasses all types of mobility while immigrant is specific to those who settled with permanent status or citizenship in a new country. That being said, we recognize that our review may not incorporate the experiences of temporary migrants, and thus we have incorporated this point into our discussion and our limitations.
Comments 10: The results section of this manuscript is compelling and well-organized, but the discussion doesn’t do enough to bridge your findings into AIP frameworks. AIP is extremely well-established as a concept, and I expect the discussion to add to the AIP conceptual model. The following literature may assist: Forsyth & Molinsky, 2021; Rosenwohl-Mack et al., 2020; Ahn, Kang, & Kwon, 2020; Bigonnesse & Chaudhury, 2020; Bigonnesse & Chaudhury, 2021; Golant, 2008, 2009, 2011, 2017, 2018; Canham et al., 2022; Zang, Loo, & Wang, 2022; Weil & Smith, 2016; Lehning et al., 2017; Versey, 2021
Response 10: We sincerely appreciate the provision of resources and agree that a more nuanced discussion about AIP is warranted. We have sought to bridge our discussion more meaningfully and hope the revisions satisfies the reviewer.
Round 2
Reviewer 3 Report
Comments and Suggestions for Authors
Thank you for your work on this manuscript draft. I believe the authors have done a sufficient job in responding to feedback.
As a reviewer, I believe this revision is fit for print. My only lasting critique is that some results seem to speak more generally to AIP for all populations (the ones that stand out to me are 1.1, 1.2, 1.3, 4.1, & 4.2). Depending on their analysis, I wonder how the authors could tweak the framing of this either in the results section itself or in the discussion-- this speaks to my previous concerns about the disconnect with existing AIP concept models.
Author Response
Comments 1: Thank you for your work on this manuscript draft. I believe the authors have done a sufficient job in responding to feedback. As a reviewer, I believe this revision is fit for print. My only lasting critique is that some results seem to speak more generally to AIP for all populations (the ones that stand out to me are 1.1, 1.2, 1.3, 4.1, & 4.2). Depending on their analysis, I wonder how the authors could tweak the framing of this either in the results section itself or in the discussion-- this speaks to my previous concerns about the disconnect with existing AIP concept models.
Response 1: Thank you for your comment, we fully agree that some aspects of our analysis apply to all populations, as AIP for immigrant older adults, like other older adults, requires attention and accommodation for mobility (through physically accessible built environments), independence, familiarity, safety, and social support. We concur with the literature that while AIP is a heterogeneous process and varies across individuals based on their social locations and intersecting identities, these aspects mustn't be neglected when addressing how older adults may age in place. As per your suggestion, we have elected to describe this more in detail in the discussion section. We hope that our revision attends to the reviewer's concerns.